# Salicylic and Jasmonic Acid Synergism during Black Knot Disease Progression in Plums

**DOI:** 10.3390/plants13020292

**Published:** 2024-01-18

**Authors:** Ranjeet Shinde, Murali-Mohan Ayyanath, Mukund Shukla, Walid El Kayal, Praveen Kumar Saxena, Jayasankar Subramanian

**Affiliations:** 1Department of Plant Agriculture, University of Guelph, Edmond C. Bovey Building, 50 Stone Road East, Guelph, ON N1G 2W1, Canada; rshinde@uoguelph.ca (R.S.); ayyanath@uoguelph.ca (M.-M.A.); mshukla@uoguelph.ca (M.S.); psaxena@uoguelph.ca (P.K.S.); 2Department of Plant Agriculture, University of Guelph, 4890 Victoria Ave N, Vineland Station, ON L0R 2E0, Canada; we21@aub.edu.lb; 3Faculty of Agricultural and Food Sciences, American University of Beirut, Riad El Solh, P.O. Box 11-0236, Beirut 1107-2020, Lebanon

**Keywords:** *Apiosporina morbosa*, black knot, phytohormones, plant growth regulators, plant immunity, plant–pathogen interactions

## Abstract

Black knot (BK) is a deadly disease of European (*Prunus domestica*) and Japanese (*Prunus salicina*) plums caused by the hemibiotrophic fungus *Apiosporina morbosa.* Generally, phytopathogens hamper the balance of primary defense phytohormones, such as salicylic acid (SA)–jasmonic acid (JA) balance, for disease progression. Thus, we quantified the important phytohormone titers in tissues of susceptible and resistant genotypes belonging to European and Japanese plums at five different time points. Our previous results suggested that auxin-cytokinins interplay driven by *A. morbosa* appeared to be vital in disease progression by hampering the plant defense system. Here, we further show that such hampering of disease progression is likely mediated by perturbance in SA, JA, and, to some extent, gibberellic acid. The results further indicate that SA and JA in plant defense are not always necessarily antagonistic as most of the studies suggest but can be different, especially in woody perennials. Together, our results suggest that the changes in phytohormone levels, especially in terms of SA and JA content due to BK infection and progression in plums, could be used as phytohormonal markers in the identification of BK-resistant cultivars.

## 1. Introduction

Plant-invading pathogens based on nutrient acquisition are classified as necrotrophs, biotrophs, and hemibiotrophs [1]. Hemibiotrophs show a biotrophic phase of lifestyle in the beginning and later enter the necrotrophic phase of lifestyle [2]. Plant defense responses often adapt to the lifestyle of infecting pathogens, with salicylic acid (SA)-dependent defense system acting against biotrophs and hemibiotrophs, while jasmonic acid (JA)- and ethylene (ET)-dependent defense systems working against necrotrophs [3,4]. Further, increased SA enhances resistance, while increased JA could make the plant susceptible to biotrophs and hemibiotrophs [5,6]. The complex defense network of plants is further manipulated by other hormones, including auxins, gibberellic acid (GA), cytokinins, ET, and abscisic acid (ABA) [7]. Pathogens also use different strategies to manipulate the plant defense system to enter and colonize the plant, thus establishing the disease [8]. A common strategy used by pathogens is the manipulation of hormone signaling to promote disease [9]. Phytopathogens exploit the plant’s defense network by either directly or indirectly producing and/or manipulating the signaling pathway(s) of plant defense system-related hormones. Many plant pathogens produce hormones that contribute to virulence, and thus, hormones are often considered virulence factors [10]

Both the European (*Prunus domestica*) and Japanese (*Prunus salicina*) plums are affected by the fungus *Apiosporina morbosa*, which causes a tumor-like disease known as ‘black knot’ (BK) [11]. Agrios [2] considered *A. morbosa* to be a hemibiotrophic fungus because of its biotrophic behavior at the early stage of infection and its conversion into a necrotrophic phage at the later stage. Warty, black knots that can range in size from half an inch to over a foot in length, are a sign of the disease [11]. Once established, the disease advances and gets worse. Old knots occasionally develop a powdery pink or white saprophytic fungal growth, and they are frequently attacked by various insects, which cause additional harm to the trees. Multiple infections cause trees to lose vigor, blossom poorly, become less productive overall, and become more vulnerable to additional harm, such as winter injury [12]. BK causes significant financial losses in North America [13,14]. Furthermore, BK has not received enough attention from the scientific community; this could be due to the challenges of testing BK under controlled conditions, unlike other fungal diseases.

The knot-forming nature of the BK disease points out the possible involvement of phytohormones such as auxins and/or cytokinins in symptom development. We have previously reported that hormones such as auxin and cytokinins seem to be highly responsible for enhancing the plum’s susceptibility to BK [15]. Conventionally, SA and JA show antagonistic behavior, with SA levels increasing against the biotrophic pathogen and JA levels increasing against the necrotrophic/hemibiotrophic pathogen. Thus, it is often portrayed that SA is involved in resistance against biotrophic pathogens, although the reasons behind this behavior are not fully clear. Here, we show that SA and JA act differently during BK disease progression than previously reported, providing a better insight into their relationship against pathogen infection, especially in perennial species. Understanding the changes in SA and JA content after BK disease infection and progression in plums might help in the identification of BK-resistant cultivars using phytohormones as markers.

## 2. Results

Susceptible genotypes had generally higher levels of both SA and JA throughout the black knot progression. The levels of SA were generally higher in susceptible genotypes of European and Japanese plums as compared to resistant ones throughout the five BK developmental stages, but the difference is statistically nonsignificant (Figure 1A,B). Jasmonic acid was significantly higher in susceptible genotypes than in resistant ones at all the stages in both types of plums except for the last stage in the Japanese plum, where the difference is not statistically significant (Figure 1C,D). Overall, the European plum possessed higher (approximately 1.3–2 fold) SA (Figure 1A,B) and JA (Figure 1C,D) titers compared with the Japanese plum. It is interesting to note that both JA and SA levels are high following BK infection, while the conventional observations always pit them as antagonistic hormones. The concomitant rise in JA and SA triggered a deeper look at other hormones.

### 2.1. Salicylic Acid

SA levels were quite high at the onset of BK (stage 1) and peaked in stage 2 in the susceptible genotypes of both plums. Thereafter, they gradually declined, barring an unexplained increase in stage 4 of Japanese plums. Interestingly, even the resistant genotypes followed the same trend, although the SA levels in resistant genotypes were always lower than in the susceptible genotypes. Only in stage 5 of European plums was very similar SA content between susceptible and resistant genotypes (Figure 1A,B). Overall, the European plum had significantly higher (1.42 times) amounts of SA than the Japanese plum (F_1,144_ = 7.66; *p* = 0.0064), and susceptible genotypes contained significantly higher (1.93-fold) amounts of SA than resistant genotypes (F_1,144_ = 25.33; *p* = 0.0001).

### 2.2. Jasmonic Acid

The European plum contained significantly higher (1.3 times) amounts of JA than the Japanese plum (F_1,142_ = 8.71; *p* = 0.0037). When the data from European and Japanese plums were analyzed together, the susceptible genotypes had significantly higher (2.21 times) amounts of JA than the resistant ones (F_1,142_ = 70.33; *p* = 0.0001). Separately, European and Japanese susceptible genotypes had 1.93 times and 2.68 times higher amounts of JA than resistant ones, respectively but the differences were statistically nonsignificant. Both in European and Japanese plums, at all BK developmental stages, JA levels were higher in susceptible genotypes than in resistant genotypes, but the difference was not statistically significant at the 2nd and 5th stages in the European plum and the 2nd and 3rd stages in the Japanese plum (Figure 1C,D). In susceptible European plums, JA levels peaked at stage 3 and then dropped until stage 5, while in susceptible Japanese plums, after a slight dip in stage 2, the JA levels increased exponentially until stage 5 (Figure 1C,D).

### 2.3. Principal Component Analysis

In PCA of susceptible genotypes of European plum, approximately 61–67% of the variability was explained by PCA I, and approximately 20–26% was explained by PCA II (Figure 2). However, in the resistant genotypes, approximately 49–67% of the variability was explained by PCAI, and 30% was explained by PCA II. In PCA of susceptible genotypes of Japanese plum, approximately 44–84% of the variability was explained by PCA I, and approximately 10–39% was explained by PCA II. However, in the resistant genotypes, approximately 70% of the variability was explained by PCA I, and 25% was explained by PCA II. At stage 1, in European susceptible genotypes, GA and BA were positively correlated, while other compounds were negatively correlated with GA and BA. However, zeatin was unrelated to any of the groups. It is noteworthy that there was a strong correlation between IAA and JA at stage 1 (>0.8). At stage 5, JA, SA, ip, and zeatin appeared to be highly correlated (>0.9). Zeatin and ip were moderately negatively correlated with GA (>0.8). Interestingly, this was not the case in resistant genotypes of the European plum. Here, at stages 1 and 5, all the compounds were positively correlated. Intriguingly, at stage 1 and stage 5 of resistant European genotypes, zeatin and ip and zeatin and SA were unrelated to other compounds, respectively. At stage 1, zeatin and ip, IAA and JA, and GA, BA, and SA were highly correlated (Figure 2). Moreover, these results are in agreement with the phytohormonal responses at specific stages (Figure 1 and Figure 2).

In Japanese susceptible plum genotypes, at stages 1 and 5, JA and SA (>0.7) and zeatin, ip, GA, and IAA were clustered (>0.8) but negatively correlated with each other (Figure 3). In the resistant Japanese genotypes, all compounds, except for BA, were highly positively correlated at stages 1 and 5 (>0.7) (Figure 3). Surprisingly, at stage 5, except for IAA, all compounds were positively correlated, whereas BA was exceptionally poorly correlated to other compounds (<0.3) (Figure 3). It is noteworthy that IAA was undetected and unrelated in both the resistant genotypes at stage 5 (Figure 1 and Figure 3).

Overall, our results suggest that reduced GA and increased auxin, cytokinins, SA, and JA appeared during BK disease progression.

## 3. Discussion

BK disease advances very slowly; the full development of the knots takes almost two years. Consistently controlled infection is a failure due to its incredibly slow nature and environment-dependent infection process. Therefore, before a phenotypic determination of resistant genotypes can be made, one must rely on the natural infection of susceptible genotypes in a genetically heterogeneous population over an extensive period of at least 7–8 years. This is possibly the most significant factor in the disease’s understudied status. We created a scoring system for disease progression based on years of observations, showing five distinct stages of BK development in plums (Figure 1a) [16]. In the current study, samples from different stages were collected as described previously, and concomitant branches from resistant genotypes were analyzed for changes in SA and JA titers to see if SA and/or JA can be a marker for identifying BK resistance in plums and to check the possibility of using SA and/or JA in priming of plum seedling/trees to control the BK disease in an environment-friendly way.

The discussion of IAA, zeatin, ip, BA, and GA titers is published elsewhere (Figure 4) [15]. The higher titers of SA and JA in the European plum as compared to the Japanese plum might be related to the hexaploidy nature of the European plum. Due to its hexaploid nature, the European plum might have multiple copies of specific genes involved in SA and JA synthesis, which might lead to the overexpression of those genes, resulting in higher titers of a specific compound in the European plum than in the Japanese plum. Moreover, in a study conducted on the *Populus* spp. as compared to the diploid *Populus* spp., 87 genes were upregulated in allotriploid *Populus* spp. Further, in the allotetraploid *Populus* spp., 259 upregulated genes were observed as compared to the diploid one [17]. Furthermore, we compared the most cultivated European and Japanese plums; the European plum is not a polyploid developed from the Japanese plum but an allohexaploid developed naturally by an interspecific cross between *Prunus cerasifera* and *Prunus spinosa* [18]. Thus, another possible reason for the different titers of SA and JA in European and Japanese plums could lie in the phylogenic difference between these two species.

### 3.1. Auxin vs. SA/JA in BK Disease Progression

IAA plays a key role in BK disease development, most probably through promoted JA, increased fungal virulence, and increased hypertrophy and hyperplasia leading to knot formation, which serves as a shelter for the fungus. Boosted auxin signaling can increase disease symptoms, the development of galls/knots and feeding sites, and/or the suppression of SA-mediated defense responses [19]. Enhanced JA levels were observed, while no significant difference in SA levels of susceptible genotypes of European and Japanese plums was observed as compared to corresponding resistant genotypes after BK infection. In biotrophic and hemibiotrophic pathogens–plant interactions, the mode of auxin action is the antagonistic behavior of auxin and SA signaling [20]. Further, recent studies showed that the virulence-promoting behavior of increased auxin involves the suppression of SA-mediated defenses [21,22]. *Fusarium oxysporum* needs auxin signaling and transport to colonize the host plant effectively, possibly through SA-dependent defense suppression [23]. Recent evidence suggests that in plant defense, SA and auxin pathways act antagonistically, while JA and auxin pathways share many similarities [24]. SA accumulation suppresses IAA and JA biosynthesis [25]. Arabidopsis NahG plants, which are unable to accumulate SA, showed 25 folds higher accumulation of JA after *Pseudomonas syringae* pv *tomato* DC3000 infection, indicating that pathogen-induced SA accumulation is responsible for JA suppression [26]. Auxin signaling showed decisive importance in activating the JA pathway after *Rice black-streaked dwarf virus* infection in rice [27]. Infection of citrus flower petals by *Colletotrichum acutatum* enhanced the accumulation of IAA and JA, supporting the synergism between IAA and JA [28]. Vinutha et al. (2020) [29] observed the synergetic effect between IAA and JA during virus infection on tomato plants and mentioned the strong interplay between biosynthesis pathways.

### 3.2. Cytokinins vs. SA/JA in BK Disease Progression

SA has a negative regulatory effect on cytokinin signaling [30]. Thus, plants accumulating high levels of SA may have reduced cytokinin content and/or signaling. Unexpectedly, in susceptible genotypes of European and Japanese plums, there was a trend of a lower amount of BA at all BK developmental stages as compared to corresponding resistant genotypes, except for the 4th stage in Japanese plums. This suggests dissimilarity between BA and other cytokinin titers, showing the inability of BK fungus in BA synthesis. The trend of increased SA levels in susceptible genotypes of European and Japanese plums might be suppressing BA synthesis in infected plum tissues, as an antagonism between SA and cytokinin synthesis was observed previously. However, increased SA levels did not suppress the zeatin and ip, as they could be synthesized by *A. morbosa* and not plum tissues.

Cytokinins enhance SA-mediated defense and expression of PR genes [31]. Some studies reported that high levels of cytokinins in plants are linked to resistance against viruses [32,33] and nematodes [34]. Exogenous application of cytokinins initiates an SA-mediated defense response, which explains the higher susceptibility of *ahk* mutants to *Pseudomonas syringae* pv. tomato and *Hyaloperonospora arabidopsidis*, such as foliar pathogens [30,35]. Moreover, cytokinins-induced pathogen resistance requires the SA pathway as well as may need additional signaling mechanisms [36].

Thus, depending on plant-pathogen interactions, cytokinins induce either SA-mediated defense or promote plant susceptibility. In contrast to cytokinin-mediated immunity, cytokinin-induced susceptibility is activated at sub-micromolar cytokinin levels [37,38]. The effect of cytokinins on plant immunity has been shown to work based on a dose-dependent manner in different pathosystems [30,39]. It was noted that lower concentrations of cytokinins help pathogen success [30]. Exogenous application of low concentrations of cytokinin BA (<1 µM) to Arabidopsis increased the establishment of oomycete *Hpa* on wild-types as compared to the mock treatment. Similarly, a moderate increase in cytokinins in wheat leaves increased the powdery mildew growth rather than resistance [40]. Based on the information available on the involvement of cytokinins in plant immunity, we propose that delayed tissue senescence due to low-to-moderate levels of cytokinins enhances biotrophic and hemibiotrophic diseases but resists necrotrophic diseases, while high levels of cytokinins inhibit biotrophic and hemibiotrophic diseases but support necrotrophs because of the activation of SA-mediated defense.

In addition, the concentration of other hormones with cytokinins should also be considered in the defense activation of plants against pathogens [41,42], as all plant hormones work together in a complex system. It is the interaction between multiple hormones that regulate the defense response and not a single hormone [38]. High levels of cytokinins in *A. morbosa* susceptible genotypes of plum could not promote SA accumulation and, thus, SA-mediated defense. This could be because it is not only cytokinins but also the complex action of several hormones, as mentioned earlier. More specifically, higher levels of IAA and JA in susceptible genotypes of plum to *A. morbosa* might be suppressing SA synthesis, as it is well known. It was noted that auxins work in an antagonistic way to cytokinins in the plant immunity system [43]. Another possible explanation for the lack of induction of SA-mediated defense in susceptible genotypes of plum is that turning on SA-mediated defense may need higher levels of cytokinins than available levels.

### 3.3. GA vs. SA/JA in BK Disease Progression

The trend of reduced levels of GA in susceptible genotypes of European and Japanese plums points out its suppression due to high levels of *A. morbosa*-induced cytokinins, especially zeatin and ip [15]. Moreover, GA promotes the degradation of DELLA proteins [44]. In contrast, stabilized DELLA proteins enhance JA signaling while attenuating SA signaling [45]. Therefore, DELLA proteins promote resistance to necrotrophs and susceptibility of biotrophs and hemibiotrophs by suppressing SA and promoting JA signaling [46]. Thus, enhanced GA might promote resistance to biotrophs and hemibiotrophs and susceptibility to necrotrophs through the degradation of DELLA proteins. Reduced GA levels in BK-infected tissues of susceptible genotypes of plum might promote stabilized DELLA proteins, contributing to enhanced JA and suppressed SA signaling, which might help in the establishment and development of the BK disease. In addition, the exogenous application of GA to *Allium sativum* plants increased resistance to the hemibiotrophic fungus *Fusarium verticillioides* [47], probably through the activation of SA-mediated defense. The necrotrophic fungus *Gibberella fujikuroi*, which causes the foolish-seedling disease of rice, synthesizes GA to promote disease may be through the degradation of DELLA proteins to suppress JA-mediated resistance to necrotrophs [46] and promote SA-mediated HR and thus, cell death.

### 3.4. JA and SA in BK Disease Progression

Generally, SA is involved in resistance against biotrophs and hemibiotrophs, while JA is involved in resistance against necrotrophs [48]. Often, these two hormones work antagonistically in response to a specific pathogen, with the induction of one leading to the suppression of the other [45]. However, the contradiction between JA and SA is not conserved in plants [49]. So far, several studies have shown the synergetic or neutral relationship between SA and JA pathways in several plant species [50,51,52,53,54,55]. However, unfortunately, the synergistic action of these two hormones has not received enough attention from the scientific community. It was reported that the synergistic action of SA and JA occurs in multiple genotypes of woody perennial *Populus* spp. after infection with the biotrophic rust fungus *Melampsora larici-populina* [55]. Exogenously applied SA and SA hyperaccumulating lines enhanced JA levels, and exogenously applied JA increased SA accumulation in Poplar [55]. The accumulation of SA was followed by JA after the infection of hemibiotrophic fungus *Ophiostoma novo-ulmi* to woody perennial *Ulmus americana* tissues, disclosing a synergistic action between SA and JA in woody trees [56]. In the same study, the exogenous application of SA enhanced the resistance of *U. americana* to *O. novo-ulmi*, suggesting the role of SA in resistance to hemibiotrophic pathogens. In woody plants, SA and JA pathways are not necessarily antagonistic [55]. Furthermore, woody perennials can store a large reserve of carbon and, thus, might have evolved the SA/JA-mediated co-defense system, especially since such long-living plants are subjected to simultaneous attack by multiple insects, pathogens, and herbivores. In contrast, short-living plants have a limited reserve of carbon to defend themselves against multiple attackers; thus, their defense system is designed to turn on against a single attacker at a time [55]. In our study, we observed the synergism between JA and SA after the infection of European and Japanese plums with *A. morbosa*, a hemibiotrophic fungus. Moreover, an increase in cytokinins, such as ip and zeatin, might have contributed to the increase in SA as well, as mentioned earlier. However, the increase in SA was nonsignificant, unlike the increase in JA. A nonsignificant increase in SA could not promote resistance and restrict *A. morbosa* infection in plums, although enhanced SA is well known to induce resistance against hemibiotrophs. The inability of SA to induce resistance to BK might be related to the insufficient difference in SA content before and after the pathogen attack, as SA levels are already high in plums. It is crucial to understand that, in plants such as Arabidopsis and Tobacco, high levels of SA are produced immediately after biotrophic and hemibiotrophic pathogen attacks. However, similarly to plums, in plants such as rice, SA levels are already high and did not increase significantly after the pathogen attack, making the plant susceptible to hemibiotrophs, such as *Magnaporthe oryzae* [57]. Thus, the noted resistance/susceptibility differences to different pathogens in different plants might be related to the differences in SA content and signaling [38].

Similar to the antagonism between SA and JA, the conflict between auxin and SA and the collegial between auxin and JA is well known [24]. As an example, auxin-dependent suppression of SA-mediated defense and activation of JA-mediated defense were observed after infection of Arabidopsis with a hemibiotrophic fungus, *Fusarium oxysporum* [23]. *A. morbosa*-driven increase in IAA might be responsible for the hike in JA after the pathogen attack in plums. Furthermore, as mentioned earlier, reduced GA through the stabilization of DELLA proteins could have contributed to the enhanced JA content. Taken together, these results indicate that significantly promoted JA might be responsible for the increased susceptibility of European and Japanese plums to *A. morbosa*. Additionally, in Cucumber Mosaic Virus-infected Arabidopsis, it has been noted that auxin and SA systemically co-increased, indicating that antagonism between SA and auxin cannot be anticipated in all the cases [58]. Similarly, we observed the co-increase in IAA and SA, supporting the fact that antagonism between SA and auxin is not conserved. Based on all these evidence we have proposed a working model on the involvement of various growth regulators on the incidence and progression of black knot in plums (Figure 5).

The interplay between JA and SA levels in black knot-infected plants reflects the complex and dynamic nature of the plant’s immune response. High JA levels in susceptible genotypes compared with resistant genotypes suggest that many regulatory features of SA–JA crosstalk play diverse and potentially ancient roles in the cell. Understanding the balance between these two signaling pathways is essential for unraveling the intricate mechanisms underlying the plant’s ability to resist and combat black knot infection. While SA induction frequently suppresses JA induction, and plants have long been hypothesized to prioritize SA over JA induction, there are seven species in which JA responses were associated with the suppression of SA induction [59]. Stroud et al. [60] suggested that the various roles of SA and JA in plants may have shaped the evolution of signaling networks. The difference in JA and SA levels could be due to genetic variations among genotypes, as suggested in this study. Some genotypes may inherently exhibit a stronger response through one pathway compared with others. Evidence from several distantly related plant species suggests that evolutionarily conserved SA- and JA-signaling crosstalk results in reciprocal antagonism between the SA and JA signaling pathways [3].

*Botrytis cinerea* secretes a virulence factorβ-(1,3)(1,6)-D-glucan, which activates the SA pathway that works antagonistically to the JA pathway, and, thus, increases disease severity in tomato. Therefore, *B. cinerea* manipulates the SA pathway to establish disease in tomato [61]. In contrast, the hemibiotrophic fungus *Ustilago maydis* secretes Cmu1 effectors to inhibit SA biosynthesis, suppressing SA-mediated immunity [62]. In addition, Topless proteins are involved in the suppression of SA signaling [63]. A few of the Tips, such as Tip1 and Tip2, and other effectors, such as Jsi1 and Nkd1 [64], induce strong SA-related defense responses, indicating antagonism between Topless proteins and effectors in SA-induced defense. Likewise, *A. morbosa* might be driving auxin to suppress SA-mediated plant resistance and promote JA-mediated susceptibility.

## 4. Materials and Methods

### 4.1. Sample Collection

From the University of Guelph’s plum breeding program, 15-year-old European (*P. domestica*) and Japanese (*P. salicina*) plum trees were selected for the current study. The phytohormonal analysis was performed using the susceptible genotypes ‘Vision’ and ‘Veeblue’ from the European plum and ‘Vampire’ and ‘Shiro’ from the Japanese plum, as well as resistant genotypes V982014 and V911415 from the European plum and ‘Underwood’ and ‘Redcoat’ from the Japanese plum to the BK disease. The resistance and susceptibility of these genotypes were established based on the assessment of BK incidence and progression over 10 years, with limited disease control in a diverse population consisting of ~150 genotypes of each Japanese and European plum (Shinde et al. Submitted). Samples from black knots at five different developmental stages (based on the stage of the infection, from the end of May till mid-October on approximately three-week intervals) were collected from susceptible genotypes and stored at −80 °C after flash freezing them using liquid nitrogen (Figure 1a). The disease starts as a swelling on the wood in early spring, bulging into a brown, lumpy swelling as the season progresses. After this, the bark ruptures, and the fungus grows into a velvety green mat on the swelling. During late summer and early fall, the green velvet is replaced by a black, cankerous layer, hence the name black knot [11,15]. In the case of resistant genotypes, at the same time points mentioned above, the branches of comparable age and size were used for the sample collection.

### 4.2. Freeze Drying and Grinding

Freeze drying of all the woody samples was done for 48 h using FreeZone 4.5 L −50 °C Benchtop Freeze Dryer (Labconco Corporation, Kansas City, MO, USA) immediately after the sample collection from the field. Then, the freeze-dried samples were stored at −80 °C until they were ground using the IKA^®^ A 11 basic Analytical mill (IKA Works, Inc., Wilmington, NC, USA) with liquid nitrogen.

### 4.3. Hormone Extraction, Identification, and Quantification

SA and JA were extracted from ground woody samples using the methanol double extraction method. Briefly, 100 mg of freeze-dried, powdered woody samples were extracted using a solvent mixture (methanol: formic acid: milli-Q water = 15:1:4). Samples were then held at −20 °C for 30 min and spun down (15 min, 4 °C, 14,000 rpm), and the supernatant was removed. A second extraction was performed on the same sample using similar conditions to those described above, and the supernatants were pooled. Solid phase extraction (Oasis^®^ HLB 1cc (30 mg), Waters Canada, Mississauga, ON, Canada) was deployed to concentrate the samples before eluting them in 200 μL methanol. Later, the eluant was filtered through a 0.22 μM centrifuge filter (Millipore; 1 min, 13,000 rpm). All standards were of analytical grade and purchased from Sigma Aldrich, Canada. SA and JA were separated using reverse phase liquid chromatography (ultra-performance liquid chromatography system (UPLC); LC-40D XS, Shimadzu, Japan) through the injection of a 5 μL aliquot of sample onto the Shim-pack Scepter LC column (2.1 × 50 mm, 1.9 μm; Mandel Scientific Company, Guelph, ON, Canada). Metabolites were separated using a gradient of solvents A (0.1% formic acid) and B (100% methanol), with the initial condition of 95% A (5% B) increased to 5% A (95% B) over 4 min using a curve of 0. The column temperature was 40 °C, and the flow rate was 0.2 mL/min. Metabolite peaks were identified by comparing them to the standards and quantified using a standard curve generated using a similar separation method and gradient conditions. SA and JA were detected using a single quadrupole mass spectrometer (LCMS 2020, Shimadzu, Japan) in single ion recording mode (SIR). JA and SA were detected in negative mode with a cone voltage of 10 for mass to charge (*m*/*z*) of 209 and 137, respectively. In both cases, the probe temperature was set to 250 °C with a gain of 5; the capillary voltage (negative) was set to 0.5 kV. The linear range for both compounds was 1.53 ng/mL–6.25 μg/mL.

### 4.4. Statistical Analysis

The study was performed using four biological replicates from each genotype and two genotypes from each group; thus, there were eight technical replicates from each group. Hormonal data of European and Japanese plums were analyzed together using general linear mixed models (proc GLIMMIX) in SAS v9.4 (SAS Institute Inc., Raleigh, NC, USA). Shapiro–Wilk normality tests and studentized residual plots were used to test error assumptions of variance analysis, including random, homogenous, and normal distributions of error. Outliers were removed using Lund’s test. Means were calculated using the LSMEANS statement, and significant differences between the treatments were determined using a post hoc LSD test α ≤ 0.05 and are mentioned in each figure. The methodology for indole 3 acetic acid (IAA), zeatin, 2-isopentenyladenine (ip), 6-benzylaminopurine (BA), and gibberellic acid-3 (GA) is described elsewhere [15]. The graphical presentation of these compounds was adapted from elsewhere [15]. Further, to underpin SA-JA interplay, at the 1st and 5th stages (extreme stages of BK infection), previously reported compounds along with SA and JA data were reanalyzed in Principal Component Analysis (PCA). Here, PCA was conducted separately on phytohormonal data from resistant and susceptible genotypes of European (Figure 2) and Japanese plums (Figure 3) using PROC PRINCOMP in SAS v9.4 (SAS Institute Inc., Raleigh, NC, USA).

## 5. Conclusions

BK disease establishment and development in European and Japanese plums seem to be achieved by the hemibiotrophic fungus *A. morbosa*-driven auxin (IAA) and cytokinins (zeatin and ip), which unbalance other phytohormones such as GA, JA, and SA to suppress the plant defense system, changing plant physiology, such as through the induction of knots and formation of the nutrient sink at the point of infection, and enhancing fungal virulence (Figure 4). SA, JA, and IAA content increased at the point of infection synergistically after BK fungal infection in multiple genotypes of both of the commercially cultivated plum species, indicating antagonism between SA and JA, and SA and IAA cannot be anticipated in all plant-pathogen interactions, especially those involving woody perennials. One or more than one among JA, IAA, zeatin, and ip might be responsible for the enhanced accumulation of SA in *A. morbosa*-infected plum tissues, but further research is needed to pinpoint the exact reason and phenomenon. For instance, more in-depth analyses of genes involved in SA and JA biosynthetic pathways might provide some answers on where the shift occurs, with SA not significantly increasing in susceptible genotypes. A nonsignificant increase in SA content after *A. morbosa* infection in susceptible plum trees could not elicit HR, suggesting an ample increase in the amount of SA that is needed for HR induction. As the role of SA and JA in disease resistance is well understood, priming seeds or seedlings or trees with SA or JA or both, depending on pathogen lifestyle (biotrophic, hemibiotrophic, and necrotrophic), could come up as an environmentally friendly approach to combat different diseases. Moreover, SA levels in the plums could be used as a phytohormonal marker to select cultivars in breeding for BK disease resistance, as SA levels show a distinct pattern for susceptibility to black knot infection in plums.

## Figures and Tables

**Figure 1 plants-13-00292-f001:**
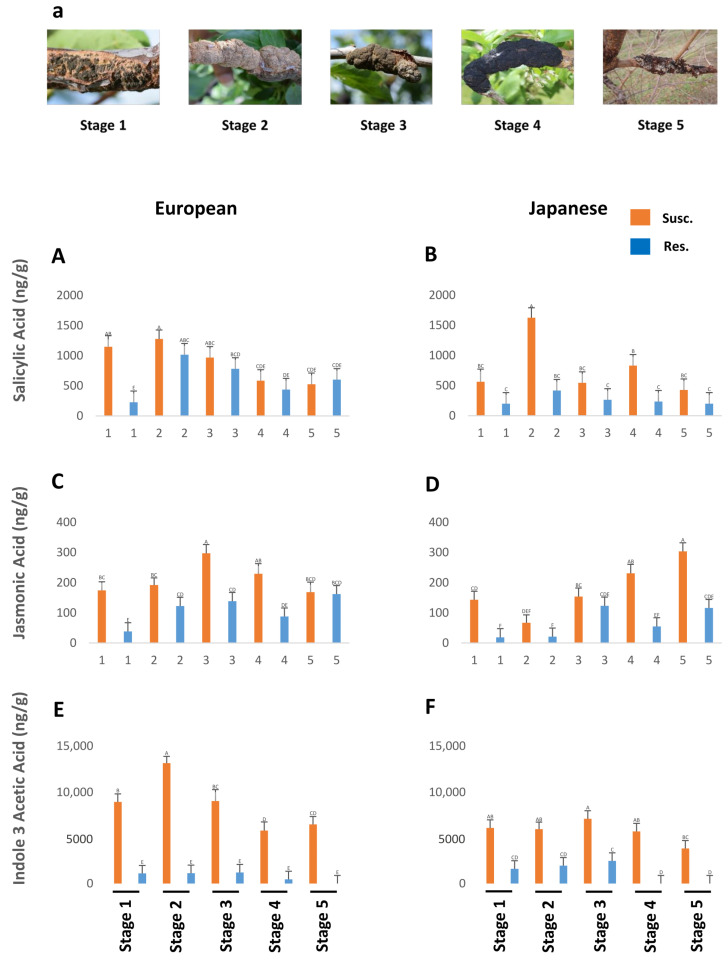
The titers of salicylic acid (*p* = 0.06) (**A**,**B**), jasmonic acid (*p* < 0.05) (**C**,**D**), and indole-3-acetic acid (*p* < 0.05) (**E**,**F**) in genotypes of European and Japanese plums resistant and susceptible to black knot (BK) at five different BK developmental stages (1–5) (**a**). Stage 1 is the appearance of visual symptoms of BK, while stage 5 is the most developed knot. Different letters denote statistical significance, and error bars represent means ± SEM (ng/g DW) for all the phytohormones.

**Figure 2 plants-13-00292-f002:**
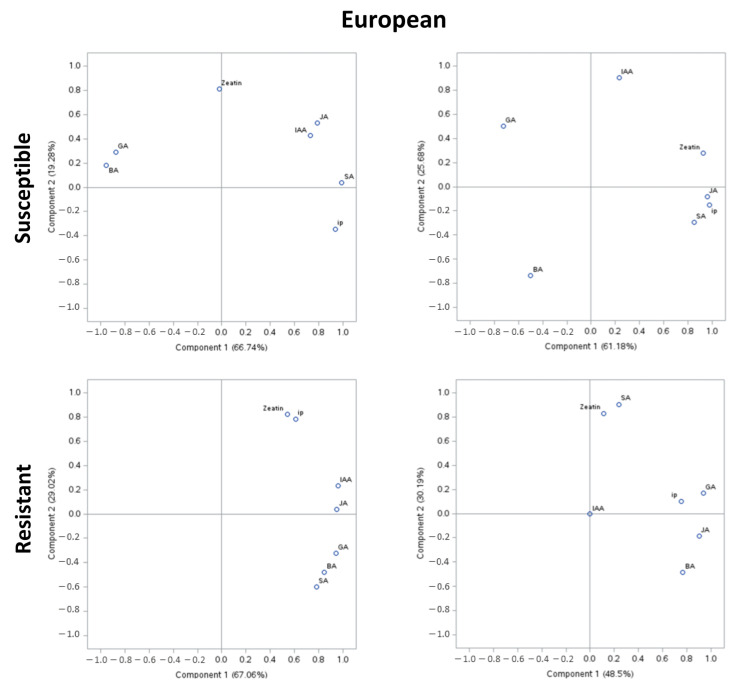
Principal component analysis (PCA) of phytohormonal contents of black knot (BK)-resistant and -susceptible genotypes of European plums at the extreme ends of BK progression, i.e., 1st (beginning of the infection) and the 5th (highly developed knot) stages. Phytohormones tryptophan (TRP), N-acetylserotonin (NAS), serotonin, indole-3-acetic acid (IAA), zeatin, 2-isopentenyladenine (ip), 6-benzylaminopurine (BA), salicylic acid (SA), jasmonic acid (JA), and gibberellic acid (GA) are labeled at data points. The distance of the data points from the center is directly proportional to the variability explained by PCA I and II, and the angle of a line passing through the center connecting two data points is inversely proportional to the correlation between phytohormones.

**Figure 3 plants-13-00292-f003:**
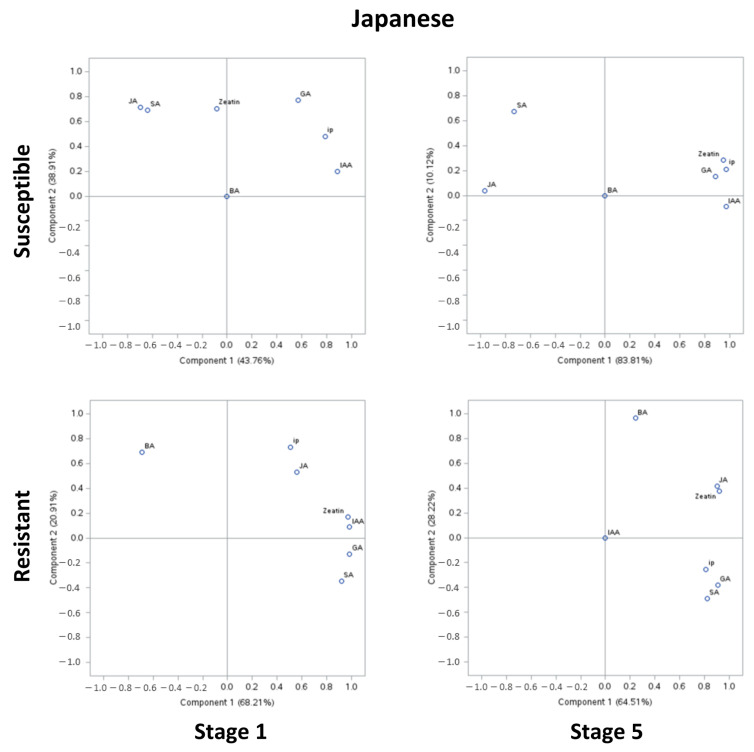
Principal component analysis (PCA) of phytohormonal contents of black knot (BK)-resistant and -susceptible genotypes of Japanese plums at the extreme ends of BK progression, i.e., 1st (beginning of the infection) and 5th (highly developed knot) stages. Phytohormones tryptophan (TRP), N-acetylserotonin (NAS), serotonin, indole-3-acetic acid (IAA), zeatin, 2-isopentenyladenine (ip), 6-benzylaminopurine (BA), salicylic acid (SA), jasmonic acid (JA), and gibberellic acid (GA) are labeled at data points. The distance of the data points from the center is directly proportional to the variability explained by PCA I and II, and the angle of a line passing through the center connecting two data points is inversely proportional to the correlation between phytohormones.

**Figure 4 plants-13-00292-f004:**
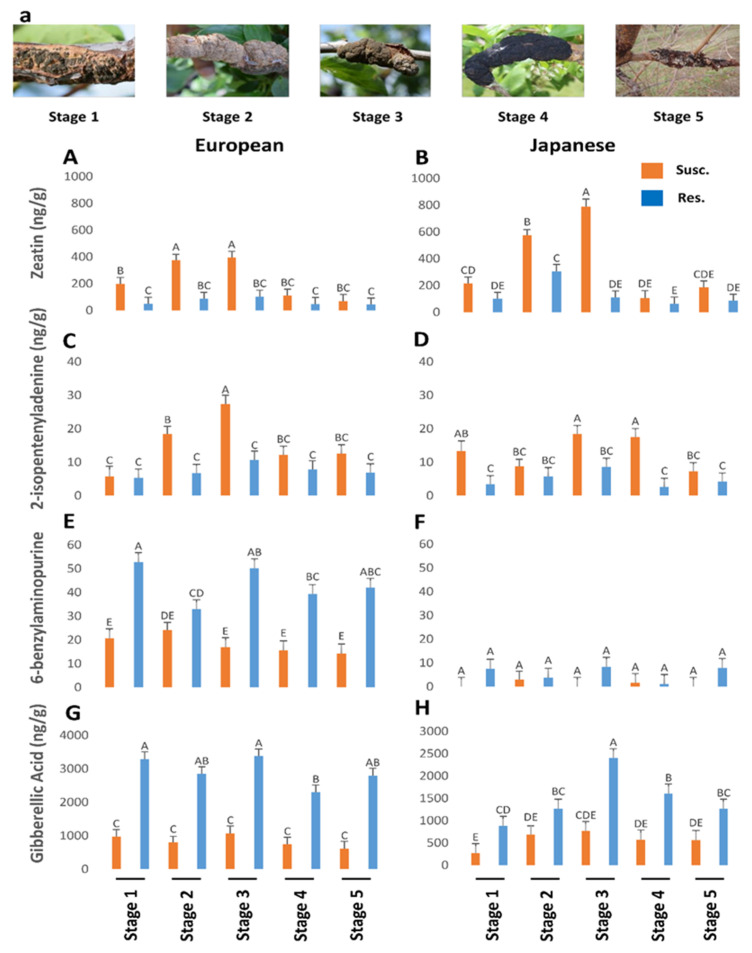
The titers of zeatin (*p* < 0.05) (**A**,**B**), 2-isopentenyladenine (p < 0.05) (**C**,**D**), 6-benzylaminopurine (*p* = 0.47) (**E**,**F**), and gibberellic acid (*p* = 0.19) (**G**,**H**) genotypes of European and Japanese plums in resistant and susceptible to black knot (BK) at five different BK developmental stages (1–5) (**a**). Stage 1 is the appearance of visual symptoms of BK, while stage 5 is the most developed knot. Different letters denote statistical significance, and error bars represent means ± SEM (ng/g DW) for all the phytohormones.

**Figure 5 plants-13-00292-f005:**
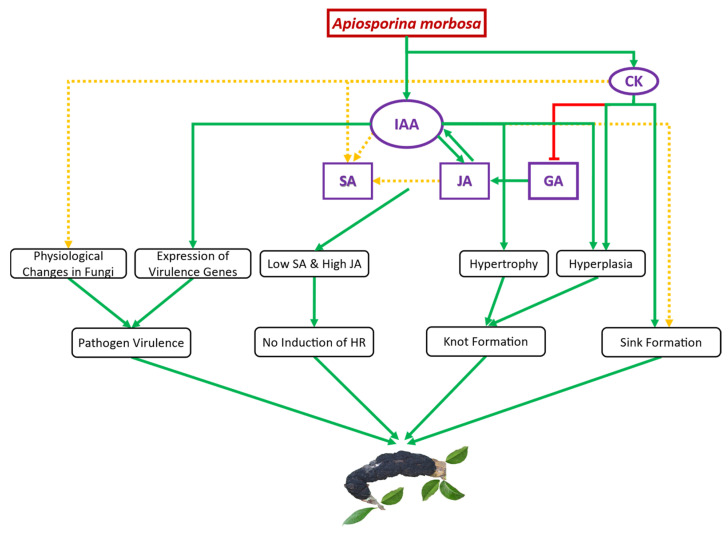
Schematic representation of the proposed auxin- and cytokinins (CK)-mediated black knot (BK) establishment and development. Immediately following infection, *Apiosporina morbosa*-driven indole-3-acetic acid (IAA) promotes jasmonic acid (JA) synthesis, and enhanced JA stimulates endogenous IAA synthesis in plum tissues. IAA, together with JA, seems to promote salicylic acid (SA) synthesis, while cytokinins may also be responsible for increased SA. Moreover, auxin causes hypertrophy and hyperplasia of infected plum cells, leading to knot formation. Cytokinin induces the suppression of gibberellic acid (GA) because elevated cytokinins might help in promoting JA synthesis in infected plum cells. In the figure, the arrow and green color indicate a positive effect, while the blunt end and red color indicate a negative effect. The solid green line denotes strong proof of disease progression, and the dotted yellow line suggests inconclusive evidence of disease progression.

## Data Availability

Data supporting the findings of this study are available within the paper. Further information can be obtained from the corresponding author.

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
