# Peer review of "Salicylic and Jasmonic Acid Synergism during Black Knot Disease Progression in Plums"

_plants, 2024, doi:10.3390/plants13020292_

Round 1

Reviewer 1 Report

Comments and Suggestions for Authors

 The research method of this paper is scientifically accurate, and the research results are fully and properly analyzed and discussed. However, the abstract part also needs to be further polished and modified to reflect the method innovation, result characteristics and conclusion significance of this paper. And figure 3 and Figure 4 the clarity of the figure need to be improved. We believe that through these efforts, this article will be more smooth, smooth, worspoken and attractive.

Comments on the Quality of English Language

The quality of English language is good, but  Minor editing of English language required.

Author Response

The research method of this paper is scientifically accurate, and the research results are fully and properly analyzed and discussed. However, the abstract part also needs to be further polished and modified to reflect the method innovation, result characteristics and conclusion significance of this paper. Rephrased the abstract as suggested.

And figure 3 and Figure 4 the clarity of the figure need to be improved.

We understand the concern, but the Figures are already according to the journal requirements. Perhaps in the final version there will be better clarity.

We believe that through these efforts, this article will be more smooth, smooth, worspoken and attractive. Thank you.

Reviewer 2 Report

Comments and Suggestions for Authors

-The introduction should include more references to changes in woody plant hormones in response to pathogens. 

-The statement of the significance of the study is not strong.

-Except for BK developmental stage 1 and 5, please give a description of characteristics in stage 2-4.

-Recommended to include discussion of why JA was increased but SA levels were not significantly different in susceptible genotypes compared to resistant genotypes?

-Lines 427-429: should be expanded to include the future prospects of this work and areas of research to be investigated, rather than only saying “further research is needed”.

-Please expand on line 435, why and how SA levels can be used as a phytohormonal marker to select BK resistance cultivars.

Comments on the Quality of English Language

The language is represented well although in some places the language is repetitive.

Author Response

Thank you, reviewer, for some insightful comments that will certainly improve the manuscript.

The introduction should include more references to changes in woody plant hormones in response to pathogens. 

We think we have included an adequate amount of references (15) in the introduction and have covered most of the important ones. If there is anything missing it would be great if the reviewer can point it out

-The statement of the significance of the study is not strong.

Added few more points to make it stronger (62-69)

-Except for BK developmental stage 1 and 5, please give a description of characteristics in stage 2-4.

Since this was described in detail in the previous references, we chose to refer that. However, as suggested, we have added a small description (370-374)

-Recommended to include discussion of why JA was increased but SA levels were not significantly different in susceptible genotypes compared to resistant genotypes?

Thank you. A small paragraph has been added to discuss this section (335-350)

-Lines 427-429: should be expanded to include the future prospects of this work and areas of research to be investigated, rather than only saying “further research is needed”.

We have added some future prospects here (439-441).

-Please expand on line 435, why and how SA levels can be used as a phytohormonal marker to select BK resistance cultivars.

Added a sentence to reflect this (448-449)